# Optimal algorithms for group distributionally robust optimization and beyond

## Abstract

Distributionally robust optimization (DRO) can improve the robustness and fairness of learning methods. In this paper, we devise stochastic algorithms for a class of DRO problems including group DRO, subpopulation fairness, and empirical conditional value at risk (CVaR) optimization. Our new algorithms achieve faster convergence rates than existing algorithms for multiple DRO settings. We also provide a new information-theoretic lower bound that implies our bounds are tight for group DRO. Empirically, too, our algorithms outperform known methods.

## 1 Introduction

Commonly, machine learning models are trained to optimize the average performance. However, such models may not perform equally well among all demographic subgroups due to a hidden bias in the training set or distribution shift in training and test phases [Hovy and Søgaard, 2015; Hashimoto *et al.*, 2018; Martinez *et al.*, 2021; Duchi and Namkoong, 2021]. Biases in datasets are also directly related to fairness concerns in machine learning [Buolamwini and Gebru, 2018; Jurgens *et al.*, 2017].

Recently, various algorithms based on distributionally robust optimization (DRO) have been proposed to address these problems [Hovy and Søgaard, 2015; Hashimoto *et al.*, 2018; Hu *et al.*, 2018; Oren *et al.*, 2019; Williamson and Menon, 2019; Sagawa *et al.*, 2020; Curi *et al.*, 2020; Zhang *et al.*, 2021; Martinez *et al.*, 2021; Duchi and Namkoong, 2021]. However, these algorithms are often highly tailored to each specific DRO formulation. Furthermore, it is often unclear whether these proposed algorithms are optimal in terms of the convergence rate. Are there a unified algorithmic methodology and a lower bound for these problems?

**Contributions.** In this paper, we study a general class of DRO problems, which includes group DRO [Hu *et al.*, 2018; Oren *et al.*, 2019; Sagawa *et al.*, 2020], subpopulation fairness [Martinez *et al.*, 2021], conditional value at risk (CVaR) optimization [Curi *et al.*, 2020], and many others. Let $\Theta \subseteq \mathbb{R}^n$ be a convex set of model parameters and $\ell(\theta; z) : \Theta \to \mathbb{R}_+$ be a convex loss of the model with parameter $\theta$ with respect to data point $z$. The data point $z$ may be drawn from one out of $m$ distributions $P_1, \ldots, P_m$ which are accessible via a stochastic oracle that returns an i.i.d. sample $z \sim P_i$. Let $Q$ be a convex subset of the probability simplex in $\mathbb{R}^m$ that contains the uniform vector, i.e., $(1/m, \ldots, 1/m) \in Q$. Our DRO formulation is as follows:

$$\min_{\theta \in \Theta} \max_{q \in Q} \sum_{i=1}^{m} q_i \mathop{\mathbf{E}}_{z \sim P_i} [\ell(\theta; z)]. \tag{1}$$

If $Q$ are the probability simplex and scaled $k$-set polytope, we can recover group DRO [Sagawa *et al.*, 2020] and subpopulation fairness [Martinez *et al.*, 2021], respectively. Moreover, we formulate a new, more general fairness concept based on weighted rankings with $Q$ being a permutahedron, which includes these special cases; see Section 2 for details.

Table 1: Summary of convergence results for group DRO. Here, $m$ denotes the number of groups, $n$ the dimension of $\theta$, $G$ the Lipschitz constant of loss function $\ell$, $D$ the diameter of feasible set $\Theta$, $M$ the range of loss function $\ell$, and $T$ the number of calls to stochastic oracle.

| reference | convergence rate $\mathbf{E}[\varepsilon_T]$ | iteration complexity | lower bound |
|---|---|---|---|
| [Sagawa *et al.*, 2020] | $O\left(m\sqrt{\frac{G^2D^2+M^2\log m}{T}}\right)$ | $O(m+n)$ + proj. onto $\Theta$ | |
| **Ours (Theorem 2)** | $O\left(\sqrt{\frac{G^2D^2+M^2m\log m}{T}}\right)$ | $O(m+n)$ + proj. onto $\Theta$ | $\Omega\left(\sqrt{\frac{G^2D^2+M^2m}{T}}\right)$ |
| **Ours (Theorem 3)** | $O\left(\sqrt{\frac{G^2D^2+M^2m}{T}}\right)$ | $O(m+n)$ + proj. onto $\Theta$ + solving scalar equation | **(Theorem 5)** |

For our general DRO, we devise an efficient stochastic gradient algorithm. Furthermore, we show that it achieves the information-theoretic optimal convergence rate for group DRO. Our main technical contributions are as follows;

- We provide a generic stochastic gradient algorithm for our general DRO. By specializing it in the group DRO setting, we provide two algorithms (GDRO-EXP3 and GDRO-TINF) that improve the rate of Sagawa *et al.* [2020] by a factor of $\Omega(\sqrt{m})$ with the almost same complexity per iteration; see Table 1. Furthermore, our generic algorithm can be specialized to improve the convergence rate of Curi *et al.* [2020] for subpopulation fairness (a.k.a. empirical CVaR optimization). Finally, we show that our algorithm runs efficiently if $Q$ is a permutahedron, which includes all aforementioned subclasses.

- We prove a matching information-theoretic lower bound for the convergence rate of group DRO. This implies that no algorithm can improve the convergence rate of GDRO-TINF (up to a constant factor). To the best of our knowledge, this is the first information-theoretic lower bound for group DRO.

- Our experiments on real-world and synthetic datasets show that our algorithms also empirically outperform the known algorithm, supporting our theoretical analysis.

## 1.1 Our techniques

**Algorithms.** The core idea of our algorithms is *stochastic no-regret dynamics* [Hazan, 2016]. We regard DRO (1) as a two-player zero-sum game between a player who picks $\theta \in \Theta$ and another player who picks $q \in Q$. The two players iteratively update their solution using online learning algorithms; in particular, we will use online gradient descent (OGD) [Zinkevich, 2003] and online mirror descent (OMD) [Cesa-Bianchi and Lugosi, 2006] for the $\theta$-player and $q$-player, respectively. In addition, we need to estimate gradients for both players, since the objective function of our DRO is stochastic and we cannot obtain exact gradients.

The convergence rate of stochastic no-regret dynamics depends on the expected regret of OGD and OMD. To obtain the optimal convergence rate, we must carefully choose the regularizer in OMD as well as gradient estimators, exploiting the structure of our DRO. In particular, we need to balance the variance of gradient estimators and the diameter terms in *both* OGD and OMD. This is the most challenging part of the algorithm design. Inspired by adversarial multi-armed bandit algorithms, we design gradient estimators for no-regret dynamics of OGD and OMD in our DRO. Indeed, our algorithms for group DRO (GDRO-EXP3 and GDRO-TINF) are based on adversarial multi-armed bandit algorithms, EXP3 [Auer *et al.*, 2003] and Tsallis-INF [Zimmert and Seldin, 2021], respectively, hence the name. Although each building block (OGD, OMD, and gradient estimators) is fairly known in the literature, we need to put them together in the right combination to obtain the optimal rate.

**Lower bound.** For the lower bound, we carefully design a family of group DRO instances for which any algorithm requires a certain number of queries to achieve a good objective value. To bound the number of queries, we use information-theoretic tools such as Le Cam's lemma and bound the Kullback-Leibler divergence between Bernoulli distributions. Such tools are also used at the heart of lower bounds for stochastic convex optimization [Agarwal *et al.*, 2012] and adversarial multi-armed bandits [Auer *et al.*, 2003], but the connection to those settings is much more subtle here, and our construction is specifically designed for group DRO-type problems.

## 1.2 Related work

DRO is a wide field ranging from robust optimization to machine learning and statistics [Goh and Sim, 2010; Bertsimas *et al.*, 2018], whose original idea dates back to Scarf [1958]. Popular choices of the uncertainty set in DRO include balls around an empirical distribution in Wasserstein distance [Esfahani and Kuhn, 2018; Blanchet *et al.*, 2019], $f$-divergence [Namkoong and Duchi, 2016; Duchi and Namkoong, 2021], $\chi^2$-divergence [Staib *et al.*, 2019], and maximum mean discrepancy [Staib and Jegelka, 2019; Kirschner *et al.*, 2020].

DRO algorithms have been mainly studied for the offline setting, i.e., algorithms can access all data points of the empirical distribution. Note that our DRO is not offline because the group distributions are given by the stochastic oracles. Namkoong and Duchi [2016] proposed stochastic gradient algorithms for offline DRO with $f$-divergence uncertainty sets. Curi *et al.* [2020] used no-regret dynamics for empirical CVaR minimization. Their algorithm invokes sampling from $k$-DPP in each iteration, which is more computationally demanding than our algorithm. Furthermore, our algorithm gets rid of an $O(\log m)$ factor in the convergence rate using the Tsallis entropy regularizer; see Theorem 4. Qi *et al.* [2021]; Jin *et al.* [2021] devised stochastic gradient algorithms for several DRO with non-convex losses.

Agarwal *et al.* [2012] gave a lower bound for stochastic convex optimization, which is a special case of our DRO with only one distribution. Recently, Carmon *et al.* [2021] showed a lower bound for minimax problem $\min_x \max_{i=1}^m f_i(x)$ for non-stochastic Lipschitz convex $f_i$. Our lower bound deals with the stochastic functions, so this result does not apply.

In this paper, we assume that the group information is given in advance. However, the group information might not be easy to define in practice. Bao *et al.* [2021] propose a simple method to define groups for classification problems based on mistakes of models in the training phase. Their method often generates group DRO instances with large $m$. Our algorithms are more efficient for such group DRO thanks to the better dependence on $m$ in the convergence rate.

No-regret dynamics is a well-studied method for solving two-player zero-sum games [Cesa-Bianchi and Lugosi, 2006]. For non-stochastic convex-concave games, one can achieve $O(1/T)$ convergence via predictable sequences [Rakhlin and Sridharan, 2013]. This result does not apply to our setting because our DRO is a stochastic game.

**Notations.** Throughout the paper, $m$ denotes the number of distributions (groups) and $n$ denotes the dimension of a variable $\theta$. For a positive integer $m$, we write $[m] := \{1, \ldots, m\}$. The orthogonal projection onto set $\Theta$ is denoted by $\mathrm{proj}_\Theta$. The $i$th standard unit vector is denoted by $\mathbf{e}_i$ and the all-one vector is denoted by $\mathbf{1}$. The probability simplex in $\mathbb{R}^m$ is denoted by $\Delta_m$.

## 2 Examples contained in our general DRO

In this section, we show how several DRO formulations in the literature can be phrased in our general DRO formulation (1). In addition, we propose a novel fairness constraint based on weighted rankings using our general DRO.

**Group DRO.** When $Q$ equals the probablility simplex, we obtain group DRO [Hu *et al.*, 2018; Oren *et al.*, 2019; Sagawa *et al.*, 2020]:

$$\min_{\theta \in \Theta} \max_{i=1}^m \mathop{\mathbf{E}}_{z \sim P_i} [\ell(\theta; z)]. \tag{2}$$

That is, group DRO aims to minimize the expected loss in the worst group, thereby ensuring better performance across all groups.

**Empirical CVaR, Subpopulation fairness, Average top-$k$ worst group loss.** Group DRO may yield overly pessimistic solutions. For instance, the groups might be automatically generated by other algorithms (such as one in Bao *et al.* [2021]) and there might exist a few "outlier" groups that make the group DRO objective trivial.

For such a case, we can restrict $Q$ to a small subset of the probability simplex so that the solution cannot put large weights on a few outlier groups. Especially, let $Q = \left\{ q \in \Delta_m : 0 \le q_i \le \frac{1}{pm} \right\}$ for

121 some parameter $p \in (0, 1)$, i.e., $Q$ is a scaled $k$-set polytope. The intuition behind the choice of $Q$
122 is that, by limiting the largest entry of $q$ to $1/pm$, DRO would optimize the expected loss over the
123 worst $p$-fraction subgroups of $m$ groups. Therefore, if the fraction of outlier groups is sufficiently
124 small compared to $p$, then $p$-fraction subgroups must contain "inlier" groups as well. Therefore, it is
125 likely that DRO with $Q$ finds solutions more robust than group DRO.

126 When $P_i$ is the Dirac measure of data $z_i$, then the resulting DRO is empirical CVaR optimization [Curi
127 *et al.*, 2020]. In the fairness context, the same problem is called subpopulation fairness [Williamson
128 and Menon, 2019; Martinez *et al.*, 2021; Duchi and Namkoong, 2021].

129 If $p = m/k$ for some positive integer $k$, the resulting DRO is the average top-$k$ worst group
130 loss [Zhang *et al.*, 2021]:

$$\min_{\theta \in \Theta} \frac{1}{k} \sum_{i=1}^{k} L_i^{\downarrow}(\theta),$$

131 where $L_i^{\downarrow}(\theta)$ denotes the the $i$th largest population group loss of $\theta$. More precisely, let $L_i(\theta) =$
132 $\mathbf{E}_{z \sim P_i}[\ell(\theta; z)]$ for $i \in [m]$ and sort them in the non-increasing order: $L_1^{\downarrow}(\theta) \geq \cdots \geq L_m^{\downarrow}(\theta)$.

133 **Weighted ranking of group losses.** The aforementioned DRO formulations are special cases of the
134 following DRO, which we call the *weighted ranking of group losses*. Let $\alpha \in \Delta^m$ be a fixed vector
135 with non-increasing entries. Let $Q$ be the permutahedron of $\alpha$, the convex hull of $(\alpha_{\sigma(1)}, \ldots, \alpha_{\sigma(m)})$
136 for all permutations $\sigma$ of $[m]$. Then, the resulting DRO is

$$\min_{\theta \in \Theta} \sum_{i=1}^{m} \alpha_i L_i^{\downarrow}(\theta).$$

137 Group DRO corresponds to $\alpha = (1, 0, \ldots, 0)$ and the average top-$k$ worst group losses corresponds
138 to $\alpha = (\underbrace{1/k, \ldots, 1/k}_{k \text{ times}}, 0, \ldots, 0)$. Another example that is contained in none of the above examples is

139 *lexicographic minimax fairness* [Diana *et al.*, 2021]. The goal of lexicographical minimax fairness
140 is to find $\theta \in \Theta$ such that the sequence $(L_1^{\downarrow}(\theta), \ldots, L_m^{\downarrow}(\theta))$ is lexicographically minimum. This
141 corresponds to $\alpha$ with sufficiently varied entries, i.e., $\alpha_1 \gg \alpha_2 \gg \cdots \gg \alpha_m$.

## 3 Algorithms

143 In this section, we describe our algorithms. First, we present a generic algorithm for our general
144 DRO (1) and provide a unified convergence analysis in Section 3.1. Then, we specialize it into two
145 concrete algorithms for group DRO (2) in Section 3.2. We sketch algorithms for the average of top-$k$
146 group losses and weighted ranking of group loss in Section 3.3.

### 3.1 Algorithm for the general case

148 We present our algorithm for geranal DRO (1). At a high level, our algorithm can be regarded as
149 stochastic no-regret dynamics. Let us denote $L(\theta, q) := \sum_{i=1}^{m} q_i \, \mathbf{E}_{z \sim P_i}[\ell(\theta; z)]$. Imagine that the
150 $\theta$-player and $q$-player run online algorithms $\mathcal{A}_\theta$ and $\mathcal{A}_q$, respectively, to solve the minimax problem
151 $\min_{\theta \in \Theta} \max_{q \in Q} L(\theta, q)$. That is, for $t = 1, \ldots, T$,

152     • $\theta_t \in \Theta$ and $q_t \in Q$ are determined by $\mathcal{A}_\theta$ and $\mathcal{A}_q$, respectively.

153     • Both players feed gradient estimators $\hat{\nabla}_{\theta,t}$ and $\hat{\nabla}_{q,t}$ to $\mathcal{A}_\theta$ and $\mathcal{A}_q$, respectively. Here,
154       $\mathbf{E}[\hat{\nabla}_{\theta,t}] = \nabla_\theta L(\theta_t, q_t)$ and $\mathbf{E}[\hat{\nabla}_{q,t}] = \nabla_q L(\theta_t, q_t)$.

155 Let

$$\varepsilon_T := \max_{q \in Q} L(\bar{\theta}_{1:T}, q) - \min_{\theta \in \Theta} \max_{q \in Q} L(\theta, q)$$

156 be the optimality gap of the averaged iterate $\bar{\theta}_{1:T} = \frac{1}{T} \sum_{t=1}^{T} \theta_t$. We can bound the expected
157 convergence rate $\mathbf{E}[\varepsilon_T]$ via regrets $R_\theta$ and $R_q$ of these online algorithms (see Appendix A for a
158 formal definition), i.e.,

$$\mathbf{E}[\varepsilon_T] \leq \frac{\mathbf{E}[R_\theta(T)] + \mathbf{E}[R_q(T)]}{T}. \tag{3}$$

159 We can obtain hence the convergence rate of the above algorithms by investigating the expected regret
160 bounds of these online algorithms.

161 To get a concrete algorithm, we must specify the online algorithms $\mathcal{A}_\theta, \mathcal{A}_q$ as well as the gradient
162 estimators $\hat{\nabla}_{\theta,t}, \hat{\nabla}_{q,t}$. We use OGD and OMD as $\mathcal{A}_\theta$ and $\mathcal{A}_q$, respectively. We construct the gradient
163 estimators by sampling $i_t \sim q_t$ and $z \sim P_{i_t}$ and setting $\hat{\nabla}_{\theta,t} = \nabla_\theta \ell(\theta_t; z)$ and $\hat{\nabla}_{q,t} = \frac{\ell(\theta_t; z)}{q_{t,i_t}} \mathbf{e}_{i_t}$.
164 This leads to Algorithm 1. There, $\Psi : Q \to \mathbb{R}$ denotes the regularizer of OMD and $\eta_{\theta,t}$ and $\eta_q$
165 denote the step sizes of OGD and OMD, respectively. [1] It turns out that this combination of online
166 algorithms and gradient estimators yields the best convergence rate (for group DRO) because the
167 expected regrets of both players are optimal.

---

**Algorithm 1** Algorithm for general DRO (1)

---

**Require:** initial solution $\theta_1 \in \Theta$, number of iterations $T$, step sizes $\eta_{\theta,t} > 0$ ($t \in [T]$), $\eta_q > 0$, and
    a strictly convex function $\Psi : Q \to \mathbb{R}$.
1: Let $q_1 = (1/m, \ldots, 1/m)$.
2: **for** $t = 1, \ldots, T$ **do**
3:    Sample $i_t \sim q_t$.
4:    Call the stochastic oracle to obtain $z \sim P_{i_t}$.
5:    $\theta_{t+1} \leftarrow \mathrm{proj}_\Theta(\theta_t - \eta_{\theta,t} \nabla_\theta \ell(\theta_t; z))$
6:    $\nabla\Psi(\tilde{q}_{t+1}) \leftarrow \nabla\Psi(q_t) - \frac{\eta_q}{q_{t,i_t}}\ell(\theta_t; z)\mathbf{e}_{i_t};$  $q_{t+1} \leftarrow \mathrm{argmin}_{q \in Q} D_\Psi(q, \tilde{q}_{t+1})$, where
    $D_\Psi(x, y) = \Psi(x) - \Psi(y) - \nabla\Psi(x)^\top(y - x)$ is the Bregman divergence with respect to $\Psi$.
7: **return** $\frac{1}{T}\sum_{t=1}^T \theta_t$.

---

168 We now analyze the convergence rate of Algorithm 1. We make the following standard assumptions.

169 **Assumption 1.** *The loss function $\ell(\theta; z)$ is continuously differentiable and $G$-Lipchitz in $\theta$, and has*
170 *range $[0, M]$ for all $z$. The Euclidean diameter of the feasible region $\Theta$ is at most $D$.*

171 The following theorem follows from plugging regret bounds of OGD and OGD, and the construction
172 of the gradient estimators into (3).

173 **Theorem 1.** *If $\eta_{\theta,t}$ is nonincreasing, Algorithm 1 achieves the expected convergence rate*

$$\mathbf{E}[\varepsilon_T] \le \frac{1}{T}\left(\frac{G^2}{2}\sum_{t=1}^T \eta_{\theta,t} + \frac{D^2}{2\eta_{\theta,T}} + \frac{M^2}{2}\eta_q \sum_{t=1}^T \mathop{\mathbf{E}}_{i_t}\left[\frac{(\nabla^2\Psi(q_t))^{-1}_{i_t,i_t}}{q_{t,i_t}^2}\right] + \frac{\max_{q^* \in Q} D_\Psi(q^*, \mathbf{1}/m)}{\eta_q}\right).$$

174 A formal proof can be found in Appendix B. We will see how specific choices of the regularizer $\Psi$
175 yield various algorithms and convergence rates for group DRO and others in the next subsections. A
176 few remarks on the regularizers, step sizes, and projection step are in order.

177 **Regularizer.** Although Algorithm 1 works with general $\Psi$, we can choose a specific regularizer for
178 $Q$ appearing in applications, e.g, the probability simplex, scaled $k$-set polytope, or a permutahedron.
179 In the next subsections, we show that the entropy regularizer $\Psi(x) = \sum_i(x_i \log x_i - x_i)$ and Tsallis
180 entropy regularizer $\Psi(x) = 2(1 - \sum_i \sqrt{x_i})$ yield efficient algorithms with improved convergence
181 rates for these cases.

182 **Step sizes.** The theorem includes decreasing step sizes such as $\eta_{\theta,t} = \frac{D}{mG\sqrt{t}}$ in addition to fixed
183 step sizes. Decreasing step sizes have the advantage that we do not require the knowledge of $T$
184 at the beginning of the algorithm but come at the cost of an extra constant factor in the expected
185 convergence rate. Since both step size policies give the asymptotically same convergence rate, we
186 describe only fixed step sizes in the theorems in the next subsections. In practice, decreasing step
187 sizes stabilize the algorithm and often outperform fixed step sizes.

188 **Projection step.** In general, the Bregman projection $\mathrm{argmin}_{q \in Q} D_\Psi(q, \tilde{q}_{t+1})$ is convex, but may
189 be costly to compute. For the applications described in Section 2, $Q$ is a permutahedron. In this case,

---

[1] We make a standard assumption that the regularizer $\Psi$ is differentiable and strictly convex, and satisfies
$\|\nabla\Psi(x)\| \to +\infty$ as $x$ tends to the boundary of $Q$.

| **Algorithm 2** GDRO-EXP3 | **Algorithm 3** GDRO-TINF |
|---|---|
| **Require:** initial solution $\theta_1 \in \Theta$, number of iterations $T$, and step sizes $\eta_{\theta,t} > 0$ ($t \in [T]$), $\eta_q > 0$. | **Require:** initial solution $\theta_1 \in \Theta$, number of iterations $T$, and step sizes $\eta_{\theta,t} > 0$ ($t \in [T]$), $\eta_q > 0$. |
| 1: Let $q_t = (1/m, \ldots, 1/m)$. | 1: Let $q_t = (1/m, \ldots, 1/m)$. |
| 2: **for** $t = 1, \ldots, T$ **do** | 2: **for** $t = 1, \ldots, T$ **do** |
| 3:   Sample $i_t \sim q_t$. | 3:   Sample $i_t \sim q_t$. |
| 4:   Call the stochastic oracle to obtain $z \sim P_{i_t}$. | 4:   Call the stochastic oracle to obtain $z \sim P_{i_t}$. |
| 5:   $\theta_{t+1} \leftarrow \mathrm{proj}_\Theta(\theta_t - \eta_{\theta,t}\nabla_\theta \ell(\theta_t; z))$ | 5:   $\theta_{t+1} \leftarrow \mathrm{proj}_\Theta(\theta_t - \eta_{\theta,t}\nabla_\theta \ell(\theta_t; z))$ |
| 6:   $\tilde{q}_{t+1} \leftarrow q_t \exp\left(\frac{\eta_q \ell(\theta_t; z)\mathbf{e}_{i_t}}{q_{t,i_t}}\right)$ | 6:   $\tilde{q}_{t+1} \leftarrow q_t \left(1 - \frac{\eta_q \sqrt{q_t}}{q_{t,i_t}}\ell(\theta_t; z)\mathbf{e}_{i_t}\right)^{-2}$ |
| 7:   $q_{t+1} \leftarrow \frac{\tilde{q}_{t+1}}{\sum_i \tilde{q}_{t+1,i}}$. | 7:   Compute $\alpha \in \mathbb{R}$ such that $\sum_{i=1}^m \left(\sqrt{\tilde{q}_{t+1,i}} - \alpha\right)^{-2} = 1$. |
| 8: **return** $\frac{1}{T}\sum_{t=1}^T \theta_t$. | 8:   $q_{t+1} \leftarrow \left(\sqrt{\tilde{q}_{t+1}} - \alpha\mathbf{1}\right)^{-2}$ |
| | 9: **return** $\frac{1}{T}\sum_{t=1}^T \theta_t$. |

it is known that the Bregman projection with respect to the entropy and Tsallis entropy regularizers can be done in $O(m \log m)$ time [Lim and Wright, 2016]. If $Q$ is the probability simplex, we even have a closed form for the Bregman projection.

## 3.2   Algorithms for Group DRO

As applications of our generic algorithm, we now describe two concrete algorithms for group DRO (2) and their convergence rates.

**GDRO-EXP3.**   Let $\Psi$ be the entropy regularizer, which corresponds to the EXP3 algorithm for the $q$-player. The resulting algorithm, GDRO-EXP3, is shown in Algorithm 2. The update is in a closed formula and its complexity is $O(m + n)$ time. The convergence rate follows from Theorem 1.

**Theorem 2.** *If $\eta_{\theta,t}$ is nonincreasing,* GDRO-EXP3 *(Algorithm 2) achieves the expected convergence rate*

$$\mathbf{E}[\varepsilon_T] \le \frac{1}{T}\left(\frac{G^2}{2}\sum_{t=1}^T \eta_{\theta,t} + \frac{D^2}{2\eta_{\theta,T}} + \frac{mM^2}{2}\eta_q T + \frac{\log m}{\eta_q}\right). \tag{4}$$

*For $\eta_{\theta,t} = \frac{D}{G\sqrt{T}}$ and $\eta_q = \sqrt{\frac{2\log m}{mM^2 T}}$, we obtain*

$$\mathbf{E}[\varepsilon_T] \le \sqrt{2}\frac{\sqrt{G^2 D^2 + 2M^2 m \log m}}{\sqrt{T}}.$$

**GRDO-TINF.**   We can further improve the convergence rate using the Tsallis entropy regularizer at the cost of a slightly higher iteration complexity. The update of $q_t$ is then

$$\tilde{q}_{t+1} = q_t \left(1 - \frac{\eta_q \sqrt{q_t}}{q_{t,i_t}}\ell(\theta_t; z)\mathbf{e}_{i_t}\right)^{-2}, \quad q_{t+1} := \left(\sqrt{\tilde{q}_{t+1}} - \alpha\mathbf{1}\right)^{-2},$$

where the multiplication, square-root, and power operations are entry-wise and $\alpha \in \mathbb{R}$ is the unique solution of equation $\sum_{i=1}^m \left(\sqrt{\tilde{q}_{t+1,i}} - \alpha\right)^{-2} = 1$. The solution $\alpha$ can be computed via the Newton method. Practically, one can use $\alpha$ in the previous iteration to warm start the Newton method. In each iteration, the algorithm performs a single orthogonal projection onto $\Theta$, the Newton method for finding $\alpha$, and $O(m + n)$ operations to update $\theta_t, q_t$. The pseudocode is given in Algorithm 3. From Theorem 1, we obtain the following convergence rate.

**Theorem 3.** *If $\eta_{\theta,t}$ is nonincreasing,* GDRO-TINF *(Algorithm 3) achieves the expected convergence rate*

$$\mathbf{E}[\varepsilon_T] \le \frac{1}{T}\left(\frac{G^2}{2}\sum_{t=1}^T \eta_{\theta,t} + \frac{D^2}{2\eta_{\theta,T}} + \sqrt{m}M^2\eta_q T + \frac{\sqrt{m}}{\eta_q}\right). \tag{5}$$

For $\eta_{\theta,t} = \frac{D}{G\sqrt{T}}$ and $\eta_q = \frac{1}{M\sqrt{T}}$, we obtain

$$\mathbf{E}[\varepsilon_T] \leq \sqrt{2}\frac{\sqrt{G^2 D^2 + 4M^2 m}}{\sqrt{T}}.$$

**Comparison to Sagawa *et al.* [2020].** Our algorithms improve the convergence rate of Sagawa *et al.* [2020] by a factor of $O(\sqrt{m})$; see Table 1. The reason lies in the choice of gradient estimator. All algorithms are stochastic no-regret dynamics. As outlined above, their convergence hence can be bounded by the regrets of the players, which depend on the variance of the local norm of the gradient estimators. Their strategy is based on uniform sampling that yields a variance of $O(m)$ for both players, whereas our bound is $O(\sqrt{m})$ thanks to the gradient estimators tailored to the regularizer of OMD. More details may be found in Appendix D.

### 3.3 Algorithm for weighted ranking of group losses

We now consider a more general case that $Q$ is a permutahedron. Applying Algorithm 1 with the Tsallis entropy regularizer, we obtain the following result.

**Theorem 4.** *If $\eta_{\theta,t}$ is nonincreasing and $Q$ is a permutahedron, Algorithm 1 with the Tsallis entropy regularizer achieves the same expected convergence rate as Theorem 3. Furthermore, the iteration complexity is $O(m \log m + n)$.*

This implies a convergence rate of $O(\sqrt{\frac{G^2 D^2 + M^2 m}{T}})$ for empirical CVaR optimization, which improves $O(\sqrt{\frac{G^2 D^2 + M^2 m \log m}{T}})$ convergence by Curi *et al.* [2020]. Furthermore, their iteration complexity is $O(m^3)$ due to the $k$-DPP sampling step, so our algorithm is even faster in terms of iteration complexity.

## 4 Lower bound

Theorem 3 states that we can find an $\varepsilon$-optimal solution for group DRO in $O(\frac{G^2 D^2 + M^2 m}{\varepsilon^2})$ calls to stochastic oracles. Next, we show that this query complexity is information-theoretically optimal.

Let $\mathcal{L}$ be a class of convex $G$-Lipschitz loss functions $\ell : \Theta \to [0, M]$. Given a loss function $\ell \in \mathcal{L}$, and an $m$-set $\mathcal{P} = \{P_1, \ldots, P_m\}$ of distributions, denote the optimality gap of $\theta \in \Theta$ by

$$R(\theta, \ell, \mathcal{P}) = \max_{P \in \mathcal{P}} \mathop{\mathbf{E}}_{z \sim P}[\ell(\theta; z)] - \min_{\theta^* \in \Theta} \max_{P \in \mathcal{P}} \mathop{\mathbf{E}}_{z \sim P}[\ell(\theta^*; z)].$$

Let $\mathcal{A}_T$ be the set of algorithms that outputs $\hat{\theta} \in \Theta$ making $T$ queries to the stochastic oracle.

**Theorem 5** (Lower Bound)**.**

$$\inf_{\hat{\theta} \in \mathcal{A}_T} \sup_{\ell \in \mathcal{L}, \Theta, \mathcal{P}} \mathbf{E}_{\mathcal{P}}[R(\hat{\theta}, \ell, \mathcal{P})] \geq \Omega\left(\max\left\{\frac{GD}{\sqrt{T}}, M\sqrt{\frac{m}{T}}\right\}\right),$$

*where $\Theta$ runs over convex sets with diameter $D$ and $\mathcal{P}$ over $m$-sets of distributions, and $\mathbf{E}_{\mathcal{P}}$ denotes the expectation over outcomes of the stochastic oracle in $\mathcal{P}$.*

As $\sqrt{x + y} \leq \sqrt{x} + \sqrt{y} \leq \sqrt{2(x + y)}$ for $x, y \geq 0$, this theorem immediately implies that the minimax convergence rate is $\Omega\left(\sqrt{\frac{G^2 D^2 + M^2 m}{T}}\right)$, which equals the convergence rate achieved by Algorithm 3 up to a constant factor.

**Proof Sketch.** It suffices to show two lower bounds $\frac{GD}{\sqrt{T}}$ and $M\sqrt{\frac{m}{T}}$ independently. The former is a well-known lower bound for stochastic convex optimization [Agarwal *et al.*, 2012]. To illustrate the latter, we take an algorithmic dependent point of view via the Le cam's method. For any algorithm in $\mathcal{A}_T$, we need to construct instances $\mathcal{P}_0, \mathcal{P}_1$ such that the total variation distance between the distributions over the query outcomes (they depend on both the behavior of the algorithm and the instance) with respect to $\mathcal{P}_0$ and $\mathcal{P}_1$ is small. On the other hand, the objective function of the two

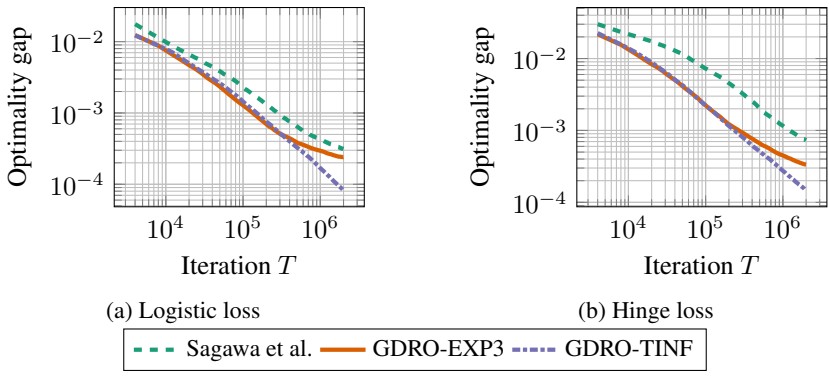

(a) Logistic loss                            (b) Hinge loss

Figure 1: Results on Adult dataset

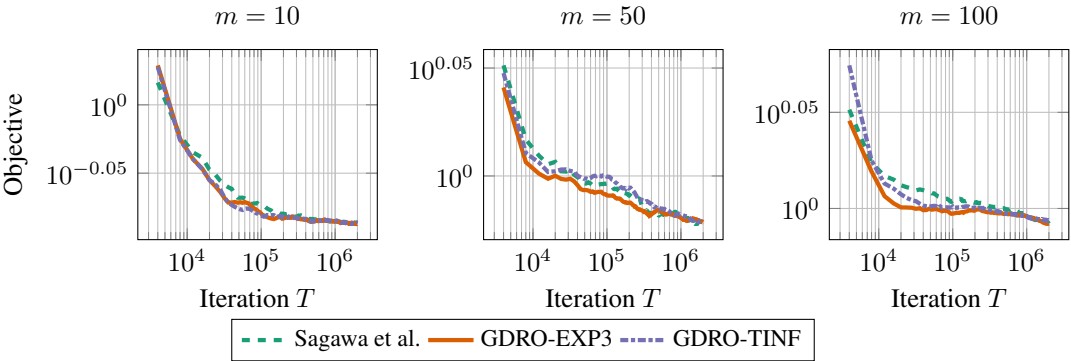

Figure 2: Results on synthetic dataset

instances must be well-separated, i.e., any fixed $\theta$ is $\delta$ sub-optimal for either $\mathcal{P}_0$ or $\mathcal{P}_1$. So, any algorithm that solves group DRO up to error $\delta$ needs to distinguish two instances $\mathcal{P}_0$ and $\mathcal{P}_1$. This implies a query lower bound because the total variation distance of the outcome distributions of these instances is small. The challenge is how to construct such instances for the regime of small dimensions of $\theta$, e.g, $n = 1$. To this end, we carefully construct linear functions for $m$ groups using opposite slopes. Then, based on the behavior of the algorithm, we tweak the noise bias in one of the groups with a positive slope, in a way that any fixed $\theta$ is $\Theta(\delta)$ sub-optimal for one of these instances. For the detailed proof, see Appendix C.

## 5 Experiments

In this section, we compare our algorithms with the known algorithm using real-world and synthetic datasets. We follow the setup in [Namkoong and Duchi, 2016].

**Adult dataset.** For the real-world dataset, we use Adult dataset [Dua and Graff, 2017]. The dataset consists of age, gender, race, educational background, and many other attributes of $48,842$ individuals from the US census. The task is to predict whether the person's income is greater than $50,000$ USD or not. We set up 6 groups based on the race and gender attributes: each group corresponds to a combination of $\{\text{black}, \text{white}, \text{others}\} \times \{\text{female}, \text{male}\}$. Converting the categorical features to dummy variables, we obtain a $101$-dimensional feature vector $a \in \mathbb{R}^n$ ($n = 101$) for each individual. We train the linear model with the logistic loss and hinge loss functions. The group-DRO objective is the worst empirical loss over the 6 groups:

$$\max_{i=1}^{6} \frac{1}{|I_i|} \sum_{(a,b) \in I_i} \ell(\theta; a, b),$$

where $I_i$ is the set of data points in the $i$th group. The feasible region is set to the Euclidean ball of radius $D = 10$.

**Synthetic dataset.** To observe the performance of the algorithms over the regime of high-dimension model parameters and the larger number of groups, we also conducted experiments using the following synthetic instances. First, we set $n = 500$ and varied $m \in \{10, 50, 100\}$. For each group $i \in [m]$, we generated the true classifier $\theta_i^* \in \mathbb{R}^n$ from the uniform distribution over the unit sphere in $\mathbb{R}^n$. The $i$th group distribution $P_i$ was the empirical distribution of 1,000 data points, where each data point $(a, b)$ was drawn as $a \sim N(0, I_n)$ and $b = \text{sign}(a^\top \theta_i^*)$ with probability $0.9$ and $b = -\text{sign}(a^\top \theta_i^*)$ with probability $0.1$. We trained the linear model with the hinge loss function. Finally, the group-DRO objective is

$$\max_{i=1}^{m} \mathbf{E}_{(a,b)\sim P_i} [\ell(\theta; a, b)].$$

The feasible region is set to the Euclidean ball of radius $D = 10$.

## 5.1 Algorithms

We implemented GDRO-EXP3, GDRO-TINF, and the algorithm in [Sagawa *et al.*, 2020] in Python. We ran our algorithms for $T = 2{,}000{,}000$ iterations.

**Inner online algorithms.** It is known that EXP3 has a variance as large as $O(T^2)$ [Lattimore and Szepesvári, 2020]. Therefore, vanilla EXP3 often fails to achieve a sublinear regret even though it achieves $O(\sqrt{T})$ regret *in expectation*. This large variance makes it difficult to reliably evaluate the performance of the algorithms. To stabilize the algorithms, we replaced EXP3 with its variation, EXP3P [Auer *et al.*, 2003], which achieves $O(\sqrt{T})$ regret *with high probability*. Note that this change does not harm our expected convergence bounds.

**Step sizes.** The choice of step sizes is crucial to the practical performance of first-order methods. We found that the decreasing step size $\eta_{\theta,t} \sim 1/\sqrt{t}$ for $\theta_t$ and the fixed step size $\eta_q \sim 1/\sqrt{T}$ for $q_t$ gave the best results. More precisely, we set $\eta_{\theta,t} = \frac{C_\theta D}{\sqrt{t}}$ ($t \in [T]$) and $\eta_q = C_q \sqrt{\frac{\log m}{mT}}$, where $C_\theta \in [0.1, 5.0]$ and $C_q \in [0.1, 3.0]$ are hyper-parameters tuned for each algorithm. We used the best hyper-parameter found by Optuna [Akiba *et al.*, 2019] for the shown results.

**Mini-batch.** The use of mini-batch often improves the stability of stochastic gradient algorithms. In our experiments, we used mini-batches of size 10 to evaluate stochastic gradients. Neither the objective values of outputs nor the stability was improved with larger mini-batch sizes. The group DRO objective is evaluated using the entire dataset.

**Initialization.** For both datasets, we initialized the algorithms with $\theta_1 = \mathbf{0}$.

## 5.2 Results

We show the results of our experiment in Figures 1 and 2.

**Adult dataset.** In Figure 1, we plot the optimality gap of the averaged iterate $\frac{1}{T} \sum_{t=1}^{T} \theta_t$ against the number of iteration $T$. We observe that all the algorithms converge with a rate roughly $T^{-0.5}$ for both loss functions, consistent with our convergence bound. Furthermore, our algorithms (GDRO-EXP3 and GDRO-TINF) achieve faster convergence compared to the algorithm by Sagawa *et al.* [2020]. Interestingly, GDRO-TINF achieves a $10^{-4}$ optimality gap in $T = 10^6$ iterations, which is faster than the theoretical $T^{-0.5}$ rate in Theorem 3.

**Synthetic dataset.** In Figure 2, we plot the objective values of the averaged iterate against the number of iterations. For all the values of $m$, our algorithms (especially GDRO-EXP3) consistently achieve smaller loss values faster than the known algorithm. The performance gap between our algorithms and the known algorithm increased as $m$ grows, which verifies that our algorithms have better dependence on $m$ in the convergence rate.

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
