# OpenReview forum: "Optimal algorithms for group distributionally robust optimization and beyond"
_NeurIPS.cc/2022/Conference — NeurIPS 2022 Submitted_

### Official Review · Reviewer_oPHP · 2022-07-11

**Rating:** 4
**Confidence:** 3
**Soundness:** 3 good
**Presentation:** 2 fair
**Contribution:** 3 good

**Summary:**

The paper proposes an algorithm for a two-player zero-sum game based on stochastic no-regret dynamics which achieves a faster convergence rate than existing algorithms for group DRO problems. They further show that their bound is tight for group DRO.


**Questions:**

# Major Comments

1) Throughout the paper formulation of "distributionally robust optimization" and "group distributionally robust optimization" terms are used interchangeably, which I find very confusing.
For example, in Line 21, the reference to general DRO problems is not precise. This paper studies the general formulation of group DRO problems because the problem introduced in (1) is already a group distributionally robust optimization problem, which is different from than traditionally studied distributionally robust optimization problem.
A similar problem occurs in Line 148, Line 143, in the title of  Algorithm 1, and so on.

To avoid confusion, the problem formulated in (1) could be called a general formulation for group distributionally robust optimization.

2) I am not entirely sure why $q_t$ is removed from the objective in the gradient update of $\theta_t$.
Is it common to apply such a technique in two-player games?
I am not sure if the unbiasedness of the gradient estimates goes through in that case.

3) I do not think the related literature for online stochastic games is covered fully. I suggest the authors to extend this part.


4) It is not clear what is $f$ in Section A.1.

5) I cannot follow the correctness of proofs because of a couple of reasons.
		- I could not see proofs for Lemma 1, 2, 3, and 4. If these lemmas are proved in a reference, please indicate them.
		- The proofs of the theorems are dependent on these lemmas.

## Minor Comments:

1) The sentence in Line 125 is neither rigorous nor clear. Please clarify this sentence.


**Limitations:**

The authors did not study the limitations of their work.


**Strengths And Weaknesses:**

The paper is well-written.

The idea seems original, however, I am not entirely sure if there is an algorithm in the multi-arm bandits that has the same convergence rate. If not this result can be seen as a contribution to the two-player stochastic online learning problems.
I highlight the need for a comprehensive literature review. Please see my comments below.

I appreciate that the authors put a separate section about the algorithm that is the most related to their work and provide a simple proof for the convergence of the competitive algorithm.

---

> ### Author Response · Authors · 2022-08-02
> **Author response to Reviewer oPHP**
>
> Thank you for your careful reading and valuable suggestions. We'll reflect your comments in the final version.
>
> > The idea seems original, however, I am not entirely sure if there is an algorithm in the multi-arm bandits that has the same convergence rate. If not this result can be seen as a contribution to the two-player stochastic online learning problems.
>
> To the best of our knowledge, there is no known multi-armed bandit algorithm with the same convergence rate. Note that our stochastic two-player game formulation is different from multi-armed bandit because we also need to optimize on a continuous variable $\theta$.
>
> > Throughout the paper formulation of "distributionally robust optimization" and "group distributionally robust optimization" terms are used interchangeably, which I find very confusing. For example, in Line 21, the reference to general DRO problems is not precise. This paper studies the general formulation of group DRO problems because the problem introduced in (1) is already a group distributionally robust optimization problem, which is different from than traditionally studied distributionally robust optimization problem. A similar problem occurs in Line 148, Line 143, in the title of Algorithm 1, and so on.
> > To avoid confusion, the problem formulated in (1) could be called a general formulation for group distributionally robust optimization.
>
> We are sorry to make confusion. We will revise the text so that DRO (1) will be called "a general form of our DRO".
>
>
> > I am not entirely sure why $q_t$  is removed from the objective in the gradient update of $\theta_t$ . Is it common to apply such a technique in two-player games? I am not sure if the unbiasedness of the gradient estimates goes through in that case.
>
> The update of $\theta_t$ indeed depends on $q_t$ because data point $z$ is sampled from group distribution $P_{i_t}$, where the group index $i_t$ is sampled from $q_t$. See Lines 3–5 in Algorithm 1. The resulting gradient estimator is unbiased.
>
>
> > I do not think the related literature for online stochastic games is covered fully. I suggest the authors to extend this part.
>
> Thank you for your suggestion. We will extend the part if you would suggest any missing references.
>
>
> > It is not clear what is $f$ in Section A.1.
>
> Functions $f_t$ in Section A.1 denote the objective functions in online learning. In our convergence analysis, we set $f_t(\theta) := L(\theta, q_t)$ (resp. $f_t(q) := L(\theta_t, q)$) to obtain regret bounds of the $\theta$-player (resp. $q$-player).
>
>
> > I cannot follow the correctness of proofs because of a couple of reasons. - I could not see proofs for Lemma 1, 2, 3, and 4. If these lemmas are proved in a reference, please indicate them. - The proofs of the theorems are dependent on these lemmas.
>
> Lemma 1 can be found in standard online learning textbooks, e.g., Theorem 6.8 in [[Orabona 2021](https://arxiv.org/abs/1912.13213)]. Lemmas 2--4 immediately follows from Lemma 1 by setting $\Psi$ to the Euclidean norm, negative entropy, and Tsallis entropy, respectively.

---

### Official Review · Reviewer_xXcs · 2022-07-11

**Rating:** 5
**Confidence:** 2
**Soundness:** 3 good
**Presentation:** 3 good
**Contribution:** 3 good

**Summary:**

This paper proposes a general class of DRO that includes Group DRO, subpopulation fairness, empirical conditional value at risk optimization and others, and the stochastic gradient learning algorithms under these frameworks. As the convergence rates shown in the previous studies are improved, the proposed algorithms are efficient, and especially, it is theoretically shown that the optimal convergence rate is achieved under the Group DRO framework. This allows to superior performance over a baseline in the Adult and synthetic datasets, which are aligned with theoretical results.

**Questions:**

* Due to its methodological simplicity, it seems that the proposed algorithms have the advantage of being applicable to a variety of datasets.
* The background for adopting the synthetic dataset was to confirm the performance of the proposed algorithm in the high-dimensional model scenario. For the readers, it seems better to utilize natural network architectures rather than only linear models. (e.g., [1] used ResNet50 for image datasets and BERT for language dataset.)
* The introduction of oracle makes a difference in terms of the setting in consideration from the existing methods, so it would be good to add a discussion on this.

[1]: (Sagawa et al.) Distributionally robust neural networks for group shifts: On the importance of regularization for worst-case generalization.

**Limitations:**

Limitations (e.g., some consideration in practice) are not specified, as mentioned in the checklist.

**Strengths And Weaknesses:**

[Strengths]
* This paper is clearly written in overall.
* It is interesting to combine online learning algorithms and appropriate gradient estimators in the scenario, where the group distributions are given by the stochastic oracles.
* It differs from previous studies in that it derives algorithms and theoretical results for online learning scenarios with stochastic functions.
* The introduction of the weighted ranking of group losses (general DRO) seems natural after Group DRO and empirical CVaR optimization.
* This paper shows several theoretical results including improvement of convergence rate compared to existing studies through the proposed algorithms and corresponding convergence analysis.

[Weaknesses]
* Compared to the theoretical results, experimental analysis and setting seem to be insufficient. (e.g., Accuracy, the number of random seeds, corresponding standard deviations can also be reported in the Appendix.)
* Line 129: ‘p=m/k’ → ‘p=k/m’

---

> ### Author Response · Authors · 2022-08-02
> **Author response to Reviewer xXcs**
>
> Thank you for your careful reading and valuable suggestions. We'll reflect your comments in the final version.
>
> > The background for adopting the synthetic dataset was to confirm the performance of the proposed algorithm in the high-dimensional model scenario. For the readers, it seems better to utilize natural network architectures rather than only linear models. (e.g., [1] used ResNet50 for image datasets and BERT for language dataset.)
>
> We did not include experiments in the deep learning regime because our main focus is to devise an efficient algorithm for a general form of our DRO (1) and prove its optimality in group DRO. Nonconvex DRO such as deep learning is much more challenging and currently out of the reach of our theoretical analysis. Note that the previous work [Sagawa et al. 2020] also analyzed the convergence for convex group DRO only.
>
> > The introduction of oracle makes a difference in terms of the setting in consideration from the existing methods, so it would be good to add a discussion on this.
>
> Thank you for your remark. Our i.i.d. oracle model to group distribution is more general than the empirical group distribution model. We will add a discussion in the revised version.

---

### Official Review · Reviewer_CKHk · 2022-07-11

**Rating:** 6
**Confidence:** 3
**Soundness:** 3 good
**Presentation:** 3 good
**Contribution:** 3 good

**Summary:**

In this paper the authors introduce new algorithms to solve a set of DRO problems consisting of Group DRO, Empirical CVar and Weighted Ranking problems. Their algorithms treats the problems as a two player game between players of the minimization and maximization problems. They leverage Online Gradient Descent and Online mirror descent for each of these respectively.
They also develop DRO instances which show a lower bound the queries required for objective performance. The bound the best possible performance of any algorithm for the Group DRO problem.


**Questions:**

What is the evaluation of the methods of this paper on other real world data sets.
What is the comparison to other methods to solve DRO problems beyond those introduced in Sagawa et al.
What are practical applications of the Group DRO, Empirical CVaR and other mentioned problems. How is the performance of the algorithm on the real world instances of these problems.

**Limitations:**

The authors do not discuss any limitations of their work. Though I doubt the existence of negative societal impact, it would be interesting to see any differences in the solutions obtained by their method as compared to Sagawa. Do this focus on different aspects etc.


**Strengths And Weaknesses:**

Originality: I feel the paper is original as it provides new and innovative algorithms for existing problems.

Quality: I feel the paper is technically sound. The novel algorithms provided in the paper improve upon existing methods and these aspects are illustrated through theoretical and numerical results. However, I feel that the numerical section is quite limited with only 1 real data set. Expanding this section will significantly improve the quality of the paper.

Clarity: I think the paper is well written and clearly communicates the various assumptions, results, numerical experiments etc.

Significance: I feel the paper is important as it takes a step forward in solve practical DRO problems by identifying new algorithms for the DRO problems.

---

> ### Author Response · Authors · 2022-08-02
> **Author response to Reviewer CKHk**
>
> Thank you for your careful reading and valuable suggestions. We'll reflect your comments in the final version.
>
> > What is the evaluation of the methods of this paper on other real world data sets.
>
> We agree that it would be interesting to see the performance of our algorithms in a different real-world dataset. We believe that one would obtain similar results for other datasets because our algorithm improved the convergence rate by $O(\sqrt m)$ and that our algorithms consistently outperform the known algorithm for both real and synthetic datasets. We will add more experiments in the final version.
>
> > What is the comparison to other methods to solve DRO problems beyond those introduced in Sagawa et al.
>
> For empirical CVaR, our method achieves $O(\sqrt{(G^2D^2 + M^2m)/T})$ convergence, improving $O(\sqrt{(G^2D^2 + M^2m\log m)/T})$ convergence of [Curi et al. 2020]. For other DRO problems, there is no known algorithm for the i.i.d. oracle setting (to the best of our knowledge).
>
> >  What are practical applications of the Group DRO, Empirical CVaR and other mentioned problems. How is the performance of the algorithm on the real world instances of these problems.
>
> Group DRO, empirical CVaR, and other DRO problems have various applications in fairness and robustness in machine learning. See references in the introduction. Our experiments demonstrated that our algorithms find an optimal solution in the worst group fairness problem (i.e., group DRO) much faster than the known method. To the best of our knowledge, there is no known algorithm for other DRO problems in the i.i.d. oracle setting, so we conducted experiments in group DRO.

---

### Official Review · Reviewer_Uz3v · 2022-07-12

**Rating:** 6
**Confidence:** 3
**Soundness:** 3 good
**Presentation:** 3 good
**Contribution:** 3 good

**Summary:**

This paper studies a generalized notion of distributionally robust optimization (DRO). In particular, the mixing weights for different data distributions ($m$ groups) can belong to arbitrary subsets $Q$ of the simplex $\Delta_m$ as long as $Q$ also contains uniform mixing weights $\frac{1}{M}\cdot \mathbb{1}$. Thus, popular DRO objectives such as group-DRO, etc., are special cases of this objective.

The paper casts this objective as a two-player game and uses stochastic gradient methods to obtain convergence guarantees for the optimality gap. The players' algorithms are simple online gradient and mirror descent algorithms with appropriate regularization. For the particular case of group-DRO, their rates improve upon the best-known rate by $\sqrt{m}$. The paper also presents a novel lower bound, showing that their algorithm has an optimal convergence under their assumptions.

**Questions:**

1) The authors use mini-batching in the experiments, but the theory doesn't have any mini-batching. I would guess that under the assumptions of $G$-lipschitzness and compact parameter space, it is hard to show such a benefit theoretically? If this is true, could some other assumptions show an improvement (such as smooth, non-Lipschitz functions)? I understand this might is a more fundamental question for online convex optimization.

2)  I believe a much more extensive empirical evaluation is possible here, particularly on data sets widely used in the fairness/robustness community. Since the theoretical results are not very hard to attain and are positive, it would be good to see that the simple algorithms outperform known algorithms on a broad range of tasks. It would also be good to see these experiments on data sets where the general objective proposed in the paper makes more sense than group DRO, etc.

**Limitations:**

No apparent negative societal impacts.

**Strengths And Weaknesses:**

I like the presentation of the paper: discussion of the related work and their connections in section two, review of current convergence rates, and discussion of algorithm design and results. The result is clean and also shown to be optimal for group DRO.

My main concern is that the algorithmic insight here is just casting the objective as a two-player game and using the correct regularizer for OMD. On the one hand, the simplicity is excellent (and OCO theory itself is known to be clean and simple), but on the other, I feel that the techniques and analysis used are not novel. This is apparent in the proof of the upper bounds, which immediately follow from known convergence lemmas. The lower bound is more interesting, and the analysis idea looks more novel. This is why I give the paper a weak acceptance.

---

> ### Author Response · Authors · 2022-08-02
> **Author feedback to Reviewer Uz3v**
>
> Thank you for your careful reading and valuable suggestions. We'll reflect your comments in the final version.
>
>
> > My main concern is that the algorithmic insight here is just casting the objective as a two-player game and using the correct regularizer for OMD. On the one hand, the simplicity is excellent (and OCO theory itself is known to be clean and simple), but on the other, I feel that the techniques and analysis used are not novel. This is apparent in the proof of the upper bounds, which immediately follow from known convergence lemmas. The lower bound is more interesting, and the analysis idea looks more novel. This is why I give the paper a weak acceptance.
>
> We believe that our OCO approach is valuable because it yields an algorithm for a wide range of DRO problems in a very simple way. In fact, the previous work on group DRO [Sagawa et al. 2020] analyzed the convergence of their algorithm with more complicated results of convex analysis. Our approach significantly simplifies their convergence analysis too; see Appendix D. Furthermore, via our OCO approach, we can obtain a unified algorithm for a more general form of DRO including group DRO, empirical CVaR, and many others. Also, the unified algorithm achieves the optimal convergence rate for group DRO, which is remarkable given the simplicity of our approach.
>
>
> > The authors use mini-batching in the experiments, but the theory doesn't have any mini-batching. I would guess that under the assumptions of G-lipschitzness and compact parameter space, it is hard to show such a benefit theoretically? If this is true, could some other assumptions show an improvement (such as smooth, non-Lipschitz functions)? I understand this might is a more fundamental question for online convex optimization.
>
> Since we study expected convergence under the i.i.d. oracle for group distributions, there is no benefit from mini-batching. This is generally true for OCO as well. We are not sure if mini-batching helps in other smoothness/Lipschitzness assumptions even in OCO. Mini-batching is known to be theoretically useful for the empirical distribution (offline) setting, but it is rather a different and more restricted setting.
> On the other hand, we observed that mini-batching can stabilize algorithms' behavior. To evaluate the average performance of algorithms reliably, we used mini-batching in our experiment. We remark that the performance trend did not change even without mini-batching (but with a larger variance).
>
>
> > I believe a much more extensive empirical evaluation is possible here, particularly on data sets widely used in the fairness/robustness community. Since the theoretical results are not very hard to attain and are positive, it would be good to see that the simple algorithms outperform known algorithms on a broad range of tasks. It would also be good to see these experiments on data sets where the general objective proposed in the paper makes more sense than group DRO, etc.
>
> Thank you for your valuable suggestion. We agree that it would be interesting to see the performance of our algorithm with various datasets and/or in other DRO problems. We will add more experiments in the final version. Below let us explain why we used the current experimental setup.
>
> **On Datasets.** We used the Adult dataset for the real-world experiment because it was used in previous work of DRO [Namkoong and Duchi, 2016]. We agree that it may be interesting to see the performance on other recent fairness datasets. We believe that one would obtain similar results for other datasets because our algorithm improved the convergence rate by $O(\sqrt m)$ and that our algorithms consistently outperform the known algorithm for both real and synthetic datasets. We will add more experiments with other datasets in the final version.
>
> **On other DRO settings.** We conducted experiments in group DRO because there is a known algorithm [Sagawa et al. 2020] under the same i.i.d. oracle model. We are not aware of any other special case of DRO for which a known algorithm exists under the same i.i.d. model. There are various works in the empirical distribution (offline) setting, but it is more restricted than the i.i.d. setting.

---

> > ### Comment · Reviewer_Uz3v · 2022-08-08
> > **Response**
> >
> > I apologize for the late response.
> >
> > > We believe that our OCO approach is valuable because it yields an algorithm for a wide range of DRO problems in a very simple way. In fact, the previous work on group DRO [Sagawa et al. 2020] analyzed the convergence of their algorithm with more complicated results of convex analysis. Our approach significantly simplifies their convergence analysis too; see Appendix D. Furthermore, via our OCO approach, we can obtain a unified algorithm for a more general form of DRO including group DRO, empirical CVaR, and many others. Also, the unified algorithm achieves the optimal convergence rate for group DRO, which is remarkable given the simplicity of our approach.
> >
> > That is a fair point but doesn't answer the question of algorithmic novelty for me. As I already said, I agree that your formulation is clean and simple.
> >
> > > Since we study expected convergence under the i.i.d. oracle for group distributions, there is no benefit from mini-batching. This is generally true for OCO as well.
> >
> > I don't think the i.i.d. oracle setting masks the benefit of mini-batching. It is the bounded gradient and parameter space assumptions.
> >
> > > We are not sure if mini-batching helps in other smoothness/Lipschitzness assumptions even in OCO.
> >
> > I don't believe it helps in OCO, but it does help in OSGD-style algorithms in bandit convex optimization. For instance, see [this paper](https://arxiv.org/abs/1209.2388).
> >
> > > On the other hand, we observed that mini-batching can stabilize algorithms' behavior. To evaluate the average performance of algorithms reliably, we used mini-batching in our experiment. We remark that the performance trend did not change even without mini-batching (but with a larger variance).
> >
> > This is precisely why I think you should analyze your algorithms for a problem class where mini-batching reduces variance. I understand that this is a more fundamental question for OCO, but strongly encourage the authors to add this point in a remark.
> >
> > > Thank you for your valuable suggestion. We agree that it would be interesting to see the performance of our algorithm with various datasets and/or in other DRO problems. We will add more experiments in the final version.
> >
> > Please do. After reading the other reviewers' comments, it seems to be a shared concern. I agree that the existing experiments make sense compared to the previous work.
> >
> > Overall, I will retain my score, and **strongly encourage the authors to add more experiments**, including but not limited to the non-convex experiments in Sagawa et al., even if the setting is not covered by your theory.

---

### Meta-Review · Area_Chair_42Pc · 2022-08-23

**Recommendation:** Reject
**Confidence:** Certain

**Metareview:**

The main criticism raised by the reviewers was the unconvincing experiments. The reviewers generally liked the simplicity of the method presented the paper, but were unconvinced by the impact/utility of the results. Even though the paper focuses on the convex regime, the paper may benefit from some DL experiments. This is not an uncommon paradigm in research (e.g. Adam and other convex optimization methods with DL experiments, etc.)

There were some other comments that are worth addressing in a future version of the paper. (e.g. adding the discussion mini-batching, improved exposition).





**Award:**

No

---

### Decision · Program_Chairs · 2022-09-14

Reject